# Deformation Behavior and Microstructural Evolution during Hot Stamping of TA15 Sheets: Experimentation and Modelling

**DOI:** 10.3390/ma12020223

**Published:** 2019-01-10

**Authors:** Zhiqiang Li, Haitao Qu, Fulong Chen, Yaoqi Wang, Zinong Tan, Mateusz Kopec, Kehuan Wang, Kailun Zheng

**Affiliations:** 1Metal Forming Technology Department, AVIC Manufacturing Technology Institute, Beijing 100024, China; zqlee98@126.com (Z.L.); quhaitao526@126.com (H.Q.); cfl0815@126.com (F.C.); xiaoqigh@sina.com (Y.W.); 2Department of Mechanical Engineering, Imperial College London, London SW7 2AZ, UK; tanzinong@163.com (Z.T.); m.kopec16@imperial.ac.uk (M.K.); kehuanhit@163.com (K.W.); 3Institute of Fundamental Technological Research, Polish Academy of Sciences, Pawinskiego 5B, 02-106 Warsaw, Poland

**Keywords:** TA15, hot stamping, phase evolution, deformation, modelling

## Abstract

Near-α titanium alloys have extensive applications in high temperature structural components of aircrafts. To manufacture complex-shaped titanium alloy panel parts with desired microstructure and good properties, an innovative low-cost hot stamping process for titanium alloy was studied in this paper. Firstly, a series of hot tensile tests and Scanning Electron Microscope (SEM) observations were performed to investigate hot deformation characteristics and identify typical microstructural evolutions. The optimal forming temperature range is determined to be from 750 °C to 900 °C for hot stamping of TA15. In addition, a unified mechanisms-based material model for TA15 titanium alloy based on the softening mechanisms of recrystallization and damage was established, which enables to precisely predict stress-strain behaviors and potentially to be implemented into Finite Element (FE) simulations for designing the reasonable processing window of structural parts for the aerospace industry.

## 1. Introduction

Titanium alloys, as advanced lightweight materials, have been extensively used in the aerospace industry, due to their high strength, low density, and good corrosion resistance [1]. The poor ductility at room temperature and loss of dimensional accuracy caused by springback drives engineers to manufacture titanium alloys, especially panel components, using hot forming techniques, such as superplastic forming (SPF) [2], hot gas forming [3], incremental forming [4] and isothermal hot forming [5]. SPF of titanium alloy is a robust technology, that has been extensively investigated and used for manufacturing extremely-complex components, normally integrated with diffusion bonding [6]. To compensate the drawbacks of low productivity, high manufacture cost and over-thinning, hot gas forming technique has been developed aiming to manufacture parts at higher strain rates and improving thickness uniformity for thin-walled components [7]. Hot gas forming is an isothermal process using high-pressure gas as loading medium, and the tube blank is heated by tools. In terms of relatively simple-shaped components, isothermal hot stamping using rigid dies is also widely used enabling forming titanium alloy sheets under the optimal process conditions. However, all the processes are carried out under an elevated temperature and isothermal condition with the forming dies heated simultaneously, which are mainly restricted by huge energy cost. Therefore, an innovative, low-cost technology for stamping titanium sheets is urgently required.

To address the above limitations, a novel hot stamping process is proposed and investigated in this study. In this process, a titanium alloy sheet is initially heated and soaked at the target temperature in a furnace. Then, the sheet is quickly transferred to the press to be hot stamped according to die profile non-isothermally. Finally, the hot stamped part is quenched by the forming dies to obtain the desired microstructure and guarantee the shape accuracy. The reasoned advantages of this novel process can be summarized as follows: (1) The absence of heating dies enables to significantly reduce the production cost, (2) relatively lower interface temperature results in a lower friction coefficient contributing to material flow, (3) the microstructure evolution is controlled by the parameters of heating, forming and subsequent quenching, so the optimal design of process can contribute to obtain the desired microstructure. Considering the process is exhibits a non-isothermal feature which results in the process windows, e.g., forming temperature, speed and transfer time etc., need to be optimized and precisely controlled. Kopec et al. [8] schematically investigated formability and microstructural evolution of Ti-6Al-4V using this novel hot stamping process. Various microstructure evolution mechanisms depending on the deformation temperature., i.e., recovery, phase transformation, and recrystallization, were identified. In addition, formability was also found to be sufficient to form sound parts at heating temperatures between 750 °C and 850 °C. The performed research successfully certificated the feasibility of hot stamping of titanium alloys. To the date, investigations of hot stamping mainly focused on two-phase titanium alloys, while another wide applied near-α titanium alloy with a higher service temperature has had limited study. 

The non-isothermal feature of hot stamping process results in the deformation temperature varies with different locations and proceeding of deformation, which determines the formability and post-form microstructure. Therefore, process parameters need to be strictly designed which can be optimized via FE simulations. To improve the simulation accuracy, a robust material model of alloy under hot stamping condition enabling the modelling of related microstructural evolutions are critical. Besides the commonly known microstructural evolutions, such as recovery, recrystallization, and dislocation accumulation and annihilation, softening is extremely important for stamping sheet metals since it determines the localized thinning and even failure. Softening of titanium alloys during hot working is determined by initial microstructure and hot working conditions. Extensive investigations were carried out. Qian et al. [9] identified the dominant softening mechanism is globularization of the secondary alpha phase for Ti-6Al-4V under hot compression conditions. Yang et al. [10] studied the flow softening and ductile damage of TA15 titanium alloy with an initial lamellar-structured α phase using hot tensile tests. Globularization was identified through SEM observations also. While for TA15 with an initial microstructure of equiaxed α phase with good fatigue resistance, the globularization is limited compared with the two-phase titanium alloy. The softening mechanism is believed to be mainly caused by recrystallization [3] and damage [10]. However, an advanced material model for near-α titanium alloy with an initial equiaxed alpha phase is lacked. 

In this paper, experimental investigations of hot stamping of a near-α titanium alloy, TA15, were performed focusing on hot formability and microstructural evolution. Schematic hot tensile tests were carried out to investigate hot deformation characteristics. SEM and Electron Backscattered Diffraction (EBSD) microstructure observations were used to identify the phase transformation and grain size evolution to provide evidence for experimental results. More importantly, using the obtained stress-strain relationships, the performed research established a mechanisms-based constitutive material model for near-α titanium alloy under hot stamping condition with considerations of a series of microstructural evolutions, especially recrystallization and damage for the softening. The developed material model can be further implemented into FE simulations to determine optimal process window and provide useful guides for manufacturing structural panel parts for the aerospace industry.

## 2. Materials and Experimental Program

### 2.1. Materials and Specimen Design

The raw material was TA15 sheet with a thickness of 1.5 mm, supplied by AVIC Manufacturing Technology Institute. The initial alloy was in annealed temper with an average grain size of α and β phase of 3 µm given by the material supplier. The initial microstructures are shown in Figure 1. Tiny and finely distributed equiaxed α phase dominants with a certain amount of granular β phase. More detailed EBSD photo of as-received TA15 is given in subsequent Section 4.2. The main alloying elements and weight percentage, provided by the material supplier, are given in Table 1. Dog-bone shaped tensile specimen was machined from the as-received sheet in the rolling direction using laser cutting. The edges were ground and polished to eliminate the heating effect of machining. The design and dimensions of the specimen are shown in Figure 2.

### 2.2. Equipment

Hot tensile tests were conducted using the thermal-mechanical material simulator, Gleeble 3800, at Imperial College London (London, UK). The thermocouple was welded in the center of the specimen, as shown in Figure 3a. The pre-experimentally determined uniform temperature gauge length is 25 mm. Cooling modules can be functionally selected by altering the gas pressure to achieve different cooling rates, which contributes to the obtaining desired microstructure after hot deformation. To avoid the oxidation at elevated temperatures, the Bornitrid Boron Nitride Spray (3M EKamold TG, Germany) was used to paint on the specimen surface before testing. The whole experimental set-up is shown in Figure 3b.

The microstructure observations were performed on, S3400 SEM microscope (Hitachi, Madidenhead, UK) at Imperial College London was used. The samples were obtained from the hot stretched specimen within the uniform temperature zone. The samples were prepared by conventional metallographic procedures for titanium, including hot mounting, grinding and polishing. After the mounting, the samples were ground by a Struers^®^ polishing machine (Struers Ltd., Rotherham, UK) using 600, 800, 1200 and 4000 SiC papers. The initial polishing was carried out using Metrep^®^ Durasilk M cloth, 3 µm diamond polishing solution and water-based lubricant. The final polishing was performed using Metrep^®^ MD-Chem cloth and 0.04 µm Colloidal Silica solution. The volume fraction of coexisting phases was measured by ImageTool Software (Version 2.00, The university of Texas Health Science Center, San Antonio, TX, USA).

### 2.3. Test Program

With regard to the hot tensile tests, a series of deformation temperatures and strain rates were performed. Figure 4 shows the temperature profile of the specimen during hot tensile tests. The specimens were heated at a rate of 4 °C/s to the target temperatures. The selected heating rate is selected considering the average heating rate of conventional furnace heating and to avoid overshooting of specimen temperature. Once the specimen reached the target temperature, the specimens were uniaxially stretched immediately to failure. To guarantee the repeatability, each test was repeated at least 2 times for consistent results. A series of deformation temperatures and strain rates were utilized as given in Table 2 to provide a full matrix testing. Considering the *β* phase transformation temperature range is between 600 °C and 990 °C [9,11], therefore, the deformation temperatures were controlled within this range. The strain rates used were 0.1 s^−1^, 1 s^−1^,10 s^−1^, which are sufficient to cover the practical forming speed in the stamping industry. For simplicity, a consistent cooling module of Gleeble is utilized for obtaining the desired microstructure.

## 3. Material Modelling

In order to establish a robust physical mechanism-based material model for hot stamping TA15, involved microstructural evolutions during hot stamping need to be known and identified first. With regard to hot forming of near-*α* titanium alloys, for *α* phase, typical microstructural evolutions, e.g., dynamic recovery (DRV), dynamic recrystallization (DRX), dynamic grain growth and phase transformation were observed [3,10]. Globularization is limited as the initial equiaxed microstructure, which is neglected in this model. While for *β* phase, dynamic recovery is dominant softening mechanism, due to the high stacking fault energy [11], so the dynamic recrystallization for the phase is not considered. Based on the above microstructure characteristics, viscoplastic constitutive equations taking into account work hardening, recovery, recrystallization and grain size evolution that were proposed by Lin et al. [12], were used as the fundamental equations in this study to predict the microstructure evolution and post-form properties, and this material models can be used for guiding the production practice and optimizing the forming parameters. The establishment of this model is given as follows in details.

The general viscoplastic flow for the alloy when considering grain size effect is expressed using Equation (1):
(1)$${\dot{\varepsilon }}_{P}={\left(\frac{\sigma /\left(1-D\right)-H-k}{K}\right)}^{n_{1}}d^{-n_{2}},$$
where ${\dot{\varepsilon }}_{p}$ is the plastic strain rate, *σ* is the flow stress, *D* is the damage factor, *H* is the hardening stress, due to dislocation evolution, *k* is the initial yield stress, *d* is the grain size. *K*, *n*_1_ are temperature dependent material parameters, and *n*_2_ is a temperature independent material constant.

For the investigated alloy, *α* and *β* phase should be considered separately. The viscoplastic flow are given in Equations (2) and (3),
(2)$${\dot{\varepsilon }}_{P,\alpha }={\left(\frac{\sigma /\left(1-D\right)-H_{\alpha }-k_{\alpha }}{K_{\alpha }}\right)}^{n_{1}}d^{-n_{2}},$$
(3)$${\dot{\varepsilon }}_{P,\beta }={\left(\frac{\sigma /\left(1-D\right)-H_{\beta }-k_{\beta }}{K_{\beta }}\right)}^{n_{3}},$$
where the subscript represents the phase.

Then, the total viscoplastic flow behavior is expressed as [13]:
(4)$${\dot{\varepsilon }}_{p}={\dot{\varepsilon }}_{p,\alpha }\left(1-f_{\beta }\right)+{\dot{\varepsilon }}_{p,\beta }f_{\beta },$$
where *f_β_* represents the volume fraction of the *β* phase.

In order to characterize the phase volume fraction variation during the non-isothermal hot stamping process, Equation (5) was used to describe the evolution rate of the volume fraction of the *β* phase [14,15].
(5)$$\dot{f_{\beta }}=X{\left(\frac{f_{\beta }}{0.5}\right)}^{\alpha }{\left(1-f_{\beta }\right)}^{\gamma }\dot{T},$$
where *X*, *α*, *γ* are temperature independent material constants.

The hardening stress *H* in Equation (1) is related to dislocation density evolution, as expressed in Equations (6) and (7) for *α* and *β* phase respectively [16].
(6)$${\dot{H}}_{\alpha }=0.5B_{\alpha }{\bar{\rho }}_{\alpha }^{\left(-0.5\right)}{\dot{\bar{\rho }}}_{\alpha },$$
(7)$${\dot{H}}_{\beta }=0.5B_{\beta }{\bar{\rho }}_{\beta }^{\left(-0.5\right)}{\dot{\bar{\rho }}}_{\beta },$$
where *B* is a temperature dependent material parameter, and $\bar{\rho }$ is the normalized dislocation density, as detailed explained in Reference [11]. The subscript represents different phases.

The dislocation evolution involves dislocation accumulation and concurrent dislocation annihilation. For *α* and *β* phase, the dislocation evolutions can be expressed using Equations (8) and (9) respectively [17].
(8)$${\dot{\bar{\rho }}}_{\alpha }=t_{1}d^{n_{4}}\left(1-{\bar{\rho }}_{\alpha }\right){\dot{\varepsilon }}_{p,\alpha }-t_{2}{\bar{\rho }}_{\alpha }^{t_{3}}-t_{4}{\bar{\rho }}_{\alpha }\frac{{\dot{S}}_{\alpha }}{1-S_{\alpha }},$$
(9)$${\dot{\bar{\rho }}}_{\beta }=t_{5}\left(1-{\bar{\rho }}_{\beta }\right){\dot{\varepsilon }}_{p,\beta }-t_{6}{\bar{\rho }}_{\beta }^{t_{7}},$$
where $\bar{\rho }$ represents the normalized dislocation density. t_1_, t_2_, t_3_, t_4_, t_5_, t_6_, t_7_ are temperature independent material constants. *n*_4_ is the temperature independent material constant. *S* represents the volume fraction of recrystallization.

For the normalized dislocation density in *α* phase, the first term represents dislocation accumulation resulting from strain hardening and static recovery. It should be noted that, grain size has a great effect on dislocation density. Dislocation is more significant when the grain size is bigger. This is because when the grain size is bigger, more grain deformation and intergranular dislocation slip would happen. Therefore, the effect of grain size on dislocation evolution is introduced in the first term. The second term represents the dislocation restoration caused by dynamic recovery. The third term represents the reduction of dislocation caused by recrystallization. Titanium alloy can have two different crystal structures, due to allotropic phase transformation: Hcp *α* phase and bcc *β* phase. The volume fraction of the *α* phase decreases as temperature increases, and complete transformation is obtained above the *β* transus temperature after a sufficient soaking, and the *β* transus temperature is 990 °C [18].

During the thermomechanical processing of titanium alloys, recovery and recrystallization are two important restoration processes. Defects like dislocation, due to the introduction of plastic deformation will provide the driving force for the initiation of both recovery and recrystallization. For materials with high stacking fault energy, recovery consumes most of the dislocations and is the dominant softening mechanism. While for materials with moderate or low stacking fault energy, recrystallization may occur when the dislocation density is higher than a threshold value. This critical threshold value of dislocation density for initiating the recrystallization is generally described by Equation (10) [19],
(10)$$\rho_{cr}=C_{\mathrm{cr}}{\left[{\dot{\varepsilon }}_{P,\alpha }\exp\left(\frac{Q_{\mathrm{cr}}}{RT}\right)\right]}^{\mathrm{cr}}$$
where Q_cr_ is the activation energy, R and *T* are ideal gas constant and the absolute temperature in Kelvin. C_cr_ and cr are material constants.

When the dislocation density is higher than *ρ_cr_*, recrystallization may occur. The recrystallization rate can be expressed by Equation (11) [19],
(11)$${\dot{S}}_{\alpha }=\frac{q_{1}{\left(0.1+S\right)}^{q_{2}}\left(1-S_{\alpha }\right){\bar{\rho }}_{\alpha }^{2}}{d},$$
where q_1_, q_2_ are temperature independent material constants. Recrystallization involves the formation of fine dislocation-free grains and the subsequent growth of these new formed grains. Therefore, recrystallization will result in the evolution of grain size during hot stamping. The grain size evolution is described using Equation (12) [20],
(12)$$\dot{d}=u_{1}d^{-w_{1}}+u_{2}{\dot{\varepsilon }}_{p,\alpha }d^{-w_{2}}-u_{3}{\dot{S}}^{w_{4}}d^{w_{3}},$$
where *u*_2_, *w*_2_, *w*_3_, *w*_4_ are temperature independent material constants and *u*_1_, *w*_1_, *u*_3_ are temperature dependent material parameters. 

In Equation (12), the first term represents the grain growth during deformation, which is related to the grain boundary mobility. The second term represents the grain refinement, due to recrystallization. 

Another typical softening mechanism refers to the hot stamping process is damage accumulation, due to the micro-voids and cracks. The continuum damage mechanism is utilized in this material model using a damage factor to represents the volume fraction of voids compared to the initial virtual microstructure. The evolution of damage is given in Equation (13),
(13)$$\dot{D}=d_{1}\cdot \left(1-D\right)\cdot {\left|{\dot{\varepsilon }}_{p,\beta }-{\dot{\varepsilon }}_{p,\alpha }\right|}^{d_{2}}+d_{6}\frac{\cosh\left(d_{3}\cdot \varepsilon_{p}\right)}{{\left(1-D\right)}^{d_{4}}}\varepsilon_{p}^{d_{5}},$$
where *d*_2_ are temperature independent material constants and *d*_1_, *d*_3_, *d*_4_, *d*_5_, *d*_6_ are temperature dependent material parameters. In Equation (13), the first term represents the damage caused by void nucleation, which is the broken of the compatibility requirements at the alpha/beta interface and the second term represents the damage caused by void growth and coalescence [21].

The total flow stress can be obtained according to Hook’s law, as given in Equation (14).
(14)$$\sigma =\left(1-D\right)E\left(\varepsilon_{t}-\varepsilon_{p}\right),$$
where *ε_t_* is the total strain, *ε_p_* is the plastic strain and *E* is Young’s modulus.

Finally, a set of constitutive equations are established. The related equations of temperature-dependent material constants are followed Arrhenius equation, as expressed in the general formula in Equation (15).
(15)$$\Theta =\Theta_{0}\exp\left(\frac{Q_{\Theta }}{RT}\right),$$
where *Θ* represents the specific temperature-dependent material constant. *Q*_Θ_ is the activation energy.

## 4. Results and Discussion

### 4.1. Hot Tensile Tests

Figure 5a shows the effect of temperature on the stress-strain relationships of TA15 samples uniaxially hot stretched at a fixed strain rate of 1 s^−1^. As can be seen that, with the increasing deformation temperature, the flow stress and strain hardening decrease gradually. Figure 5b summarizes the strain-to-failure of specimens at different temperatures. The maximum strain was found at a temperature of 850 °C, greater than 1. With further increasing the temperature, i.e., 950 °C, both the flow stress and ductility decreases severely. The reason for the flow stress decreasing is mobility of dislocation occurred more easily at a higher temperature, and more alpha phase transformed to beta phase as well. While the reasons for the ductility decreasing is believed to be mainly caused by the surface oxidation of samples resulting from the failure of the oxidation resistant lubricant at 950 °C. Then, with the proceeding of deformation, micro-cracks are easily initiated on the surface, severe oxidation initiated from the surface (failure of oxidation resistant lubricant) and penetrated, resulting in the quick fracture as observed. In addition, with regard to sheet metal forming, hardening is also an important feature determining the uniformity of deformation. Although the obvious work hardening is observed at temperatures of 650 °C and 700 °C, the strain was less than 0.3, which is insufficient to deliver sufficient strain for manufacturing complex-shaped geometries. Therefore, the feasible forming temperature window is believed to be between 750 °C and 900 °C, with a ductility greater than 0.5.

### 4.2. Microstructure Observations

Figure 6 shows the fracture morphology obtained from the hot uniaxially stretched specimens at different deformation conditions. Dimple fractures are observed for all the specimens. Comparing Figure 6a–c, it is obvious that with the increasing temperature at the same strain rate, the dimple area increased significantly proving higher ductility at a higher temperature. For the higher temperature, it is easier for the nucleation and extension of dimples resulting to obtain deeper and larger dimples. Figure 6c,d shows the fractures at 800 °C with stain rate of 1 s^−1^ and 10 s^−1^ respectively. For the higher strain rate, the dimple is shallower indicating a poorer ductility. The reason can be explained by the viscoplastic feature of hot deformation, when the alloy is deformed at a higher strain rate, the deformation time is relatively short and the diffusion is insufficient. The mobility of dislocation and grain boundary is not as sufficient as that at a slower strain rate, therefore the stress is higher and the ductility is poorer. 

As schematically illustrated in Figure 4, phase transformation occurs during the thermomechanical processing of titanium alloys, which governs the ductility and microstructural evolution. Figure 7 shows the SEM morphology of post-formed TA15 specimens at different deformation temperatures with a fixed strain rate of 1 s^−1^ to identify the phase transformation. As can be seen in this figure, considering the temperature range of *β* phase transformation is between 600 °C and 990 *β* °C, hence, when the temperature is higher than the critical *β* phase transformation temperature, 600 °C, phase transformation of *α* to *β* would occur during heating & hot deformation, and the proportion of *β* phase increases with the increase of deformation temperature; after the hot deformation, the reverse phase transformation of *β* to *α* would occur during the cooling process. It should be noted that, the transformed alpha phases are the lamellar secondary alpha within the beta phase grain, the key of distinguishing *β* phase and transformed *β* phase is whether there is a lamellar secondary *α* phases generated. When the temperatures are lower, lamellar secondary *α* phases have no time to form or the quantity is less, which cannot be observed through the conventional light microscope and SEM observations.

At 700 °C, the light beta grains were elongated, which indicates the dislocation was accumulated in the matrix, and compared with the initial microstructure, the fraction of *β* phase was increasing. It can be also seen from the Figure 7b, the precipitation of *α* secondary cannot be observed. While at 800 °C, the light intergranular *β* became equiaxed, as shown in Figure 7d, which indicates more recrystallization of *α* phase occurs than that at 700 °C. Secondary *α* was also not observed at 800 °C. When the testing temperature is 900 °C, as shown in Figure 7e, more *β* was obtained in the process of heating and deformation, and it transformed to transformed *β* in the subsequent cooling process with secondary alpha precipitated inside. Figure 8 shows the EBSD comparisons of grain size between as-received material and specimen conditioned by a typical forming condition: 850 °C and 1 s^−1^. It can be seen that the deformed material had an obvious elongated grain orientation. Some new and small grains generated at the grain boundaries, which certificated that dynamic recrystallization occurred this condition. Therefore, dynamic recovery and recrystallization are two main softening mechanism in hot stamping TA15 sheets.

It is obvious that both the fraction and morphology of the two phases depend on the processing parameters. Therefore, the microstructure evolution can be controlled by the parameters of heating, forming and subsequent cooling, so the different fraction of equiaxed and lamellar microstructure can be obtained. In consideration of different service conditions, the optimal design of process can contribute to obtaining the desired microstructure.

### 4.3. Application of the Material Model

The constant values of the established material model in Section 3 can be fitted through hot tensile stress-strain curves. The initial grain size used is 3 µm according to the characterization of the as-received alloy. The numerical fitting procedure is that, using the Genetic Algorithm (GA)-based optimization method by means of Matlab to obtain a relative value range firstly for a globally optimal solution, then using the nonlinear least square method to determine the final value class by minimizing the residuals between the computed target values and experimental data. The objective function is defined as:
(16)$$f\left(\sigma \right)=\overset{n}{\underset{i=1}{\sum }}{\left(\frac{\sigma_{c}^{i}-\sigma_{e}^{i}}{\sigma_{e}^{i}}\right)}^{2},$$
where *f*(*σ*) represents the residual error in the stress, *n* represents the deformation conditions (temperature and strain rate), and $\sigma_{c}^{i}$ and $\sigma_{e}^{i}$ are the calculated and experimental stress, respectively. The material constants in the visco-plastic model should minimize the objective function, as shown in Equation (16). Figure 9 shows the comparison of the volume fraction of *β* phase between the experimental results (solid circular symbols) and computation results (solid line). The experimental values were obtained from the post-processing of SEM images in Figure 7 using ImageTool software. Good agreements can be observed. Both the trend and value of the volume fraction of *β* phase can be precisely predicted which illustrates that the established material model enables to predict the microstructural during hot stamping of TA15.

With regard to the hot stamping process, practically, the background microstructural evolutions are very complex and different to be precisely determined. The most critical outcome of material modelling is to be able to predict the macro-scale mechanical stress-strain behaviors at different forming conditions, which is a practical interest of industries for predicting the deformation of a real formed part. Figure 10 shows the comparisons of stress-strain relationships between experimental results and computational results at different forming conditions. Excellent agreements have been found between the experimental results and computational fitting. As can be seen in these figures, besides the flow stress levels and ductility variations, the softening feature can be also precisely predicted for different temperatures and strain rates. Obvious strain rate hardening feature can be predicted, with increasing the strain rate, the flow stress level significantly increases.

## 5. Conclusions

The hot deformation and microstructural evolution of TA15 under a novel hot stamping process were experimentally investigated in this paper. A physical-based material model for near-α titanium alloy with an initial equiaxed microstructure was successfully established. The results enable to provide useful guidance for the aerospace industry with the following conclusions:

(1) With the increasing temperature, the formability of TA15 became better resulted from the recovery and recrystallization. To avoid severe *β* phase transformation, due to requirements for better fatigue properties of aircraft components, and failure of oxidation resistance lubricant at very high temperatures, i.e., 950 °C, the optimal hot forming temperature window is from 750 °C to 900 °C.

(2) According to SEM and EBSD observations, when the temperature is below 750 °C, alpha grains were elongated, and strain hardening occurred because of dislocation accumulation with no obvious softening, i.e., recovery and recrystallization. From 750 to 850 °C, *β* grains became equiaxed and grew bigger; recrystallization of *α* grains started to occur. Over 850 °C, the lamellar secondary *α* precipitated and grew thicker with higher temperature.

(3) The physical-based material model, taking into account of phase transformation effect, recovery, recrystallization and damage, was established for the near-*α* titanium alloy. The developed model enables to precisely predict the deformation behaviors of TA15 under hot stamping conditions.

## Figures and Tables

**Figure 1 materials-12-00223-f001:**
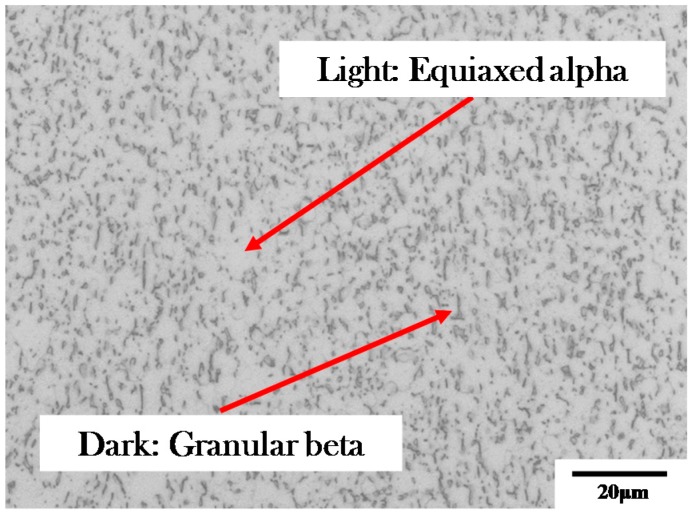
Initial microstructures of TA15 (light microscope).

**Figure 2 materials-12-00223-f002:**
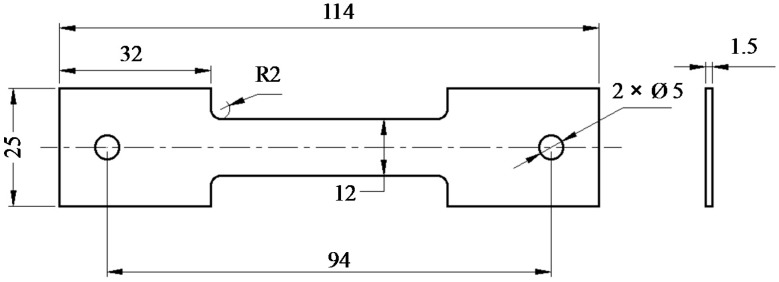
Hot tensile specimen design (all dimensions are in mm).

**Figure 3 materials-12-00223-f003:**
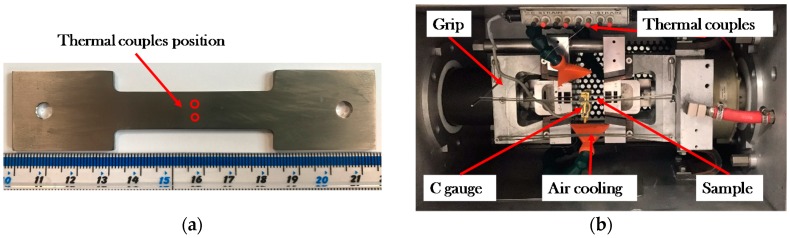
Experimental set-up (**a**) thermocouple location, and (**b**) hot tensile test set-up.

**Figure 4 materials-12-00223-f004:**
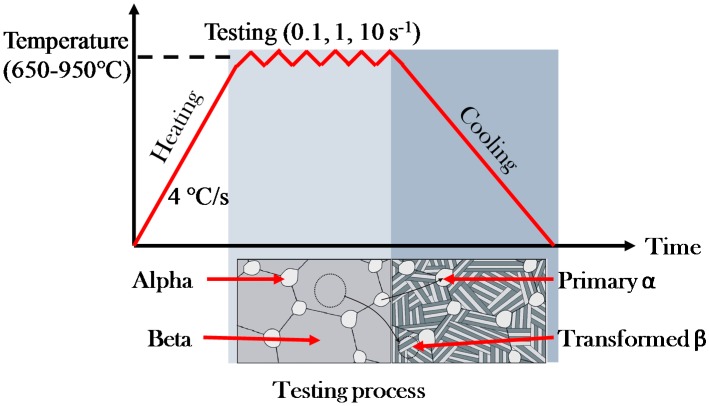
The temperature profile of the specimen during hot tensile tests.

**Figure 5 materials-12-00223-f005:**
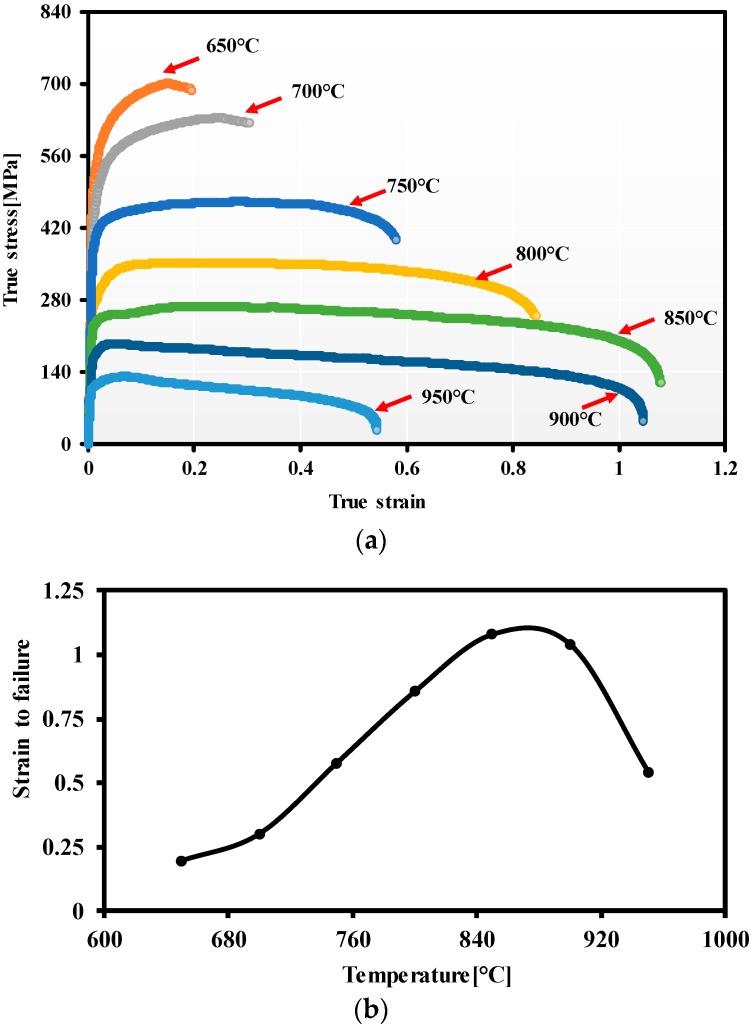
Hot tensile test results: (**a**) Stress-strain relationships and (**b**) variations of alloy ductility with temperature.

**Figure 6 materials-12-00223-f006:**
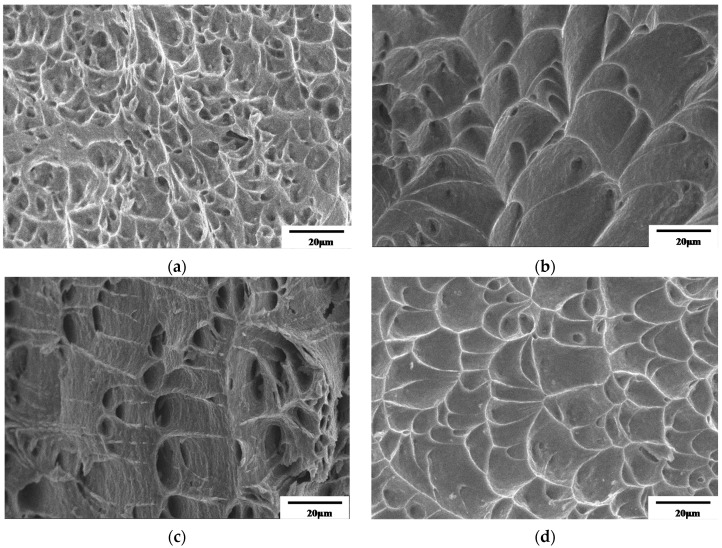
Fracture morphology obtained for TA15 at different forming conditions: (**a**) 700 °C, 1 s^−1^; (**b**) 750 °C, 1 s^−1^; (**c**) 800 °C, 1 s^−1^; (**d**) 800 °C, 10 s^−1^.

**Figure 7 materials-12-00223-f007:**
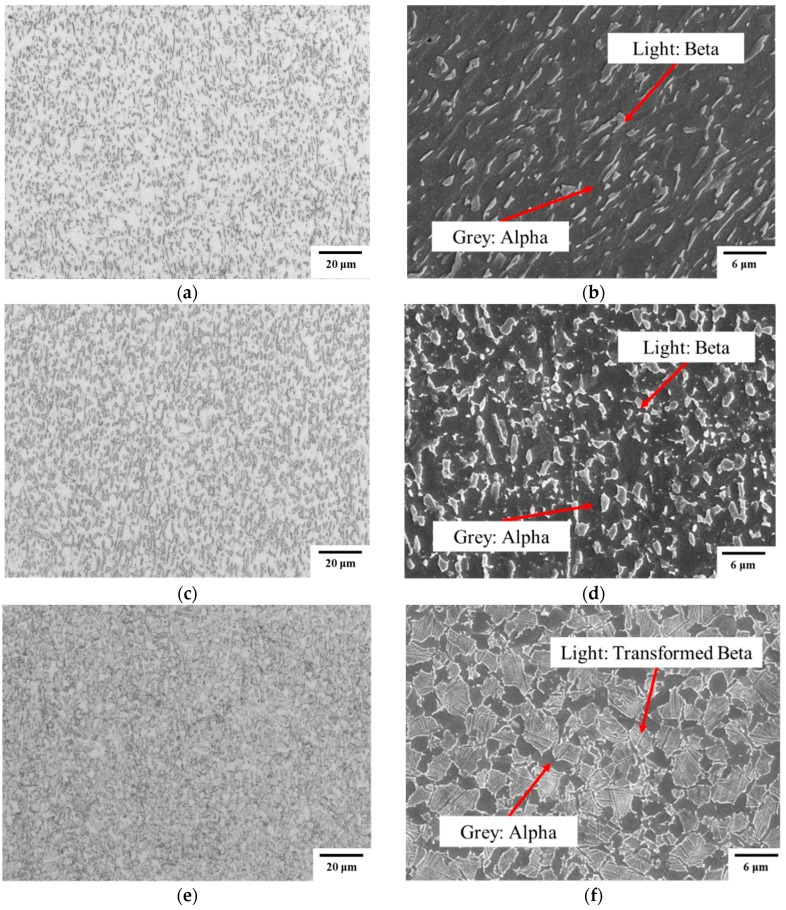
The microstructure of TA15 conditioned at constant strain rate 1 s^−1^: (**a**,**b**) 700 °C; (**c**,**d**) 800 °C and (**e**,**f**) 900 °C, after hot deformation.

**Figure 8 materials-12-00223-f008:**
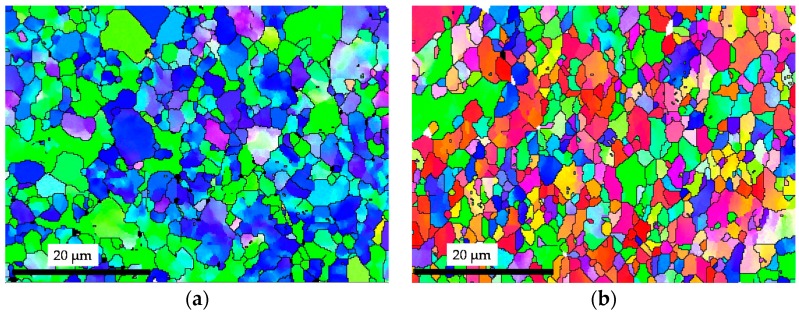
EBSD observations of dynamic recrystallization of TA15 under hot stamping condition: (**a**) As-received condition and (**b**) 850 °C and 1 s^−1^.

**Figure 9 materials-12-00223-f009:**
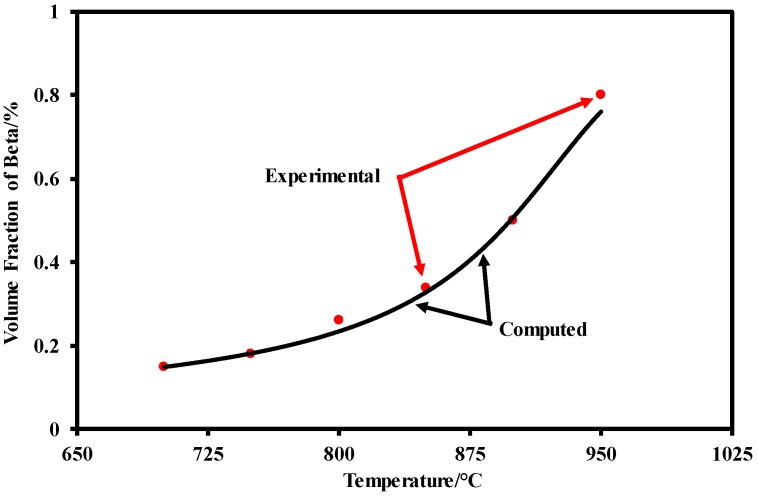
Comparison of volume fraction *β* phase evolution between experimental (symbols) and computed (solid curves) strain-stress during hot stamping TA15 at different temperatures.

**Figure 10 materials-12-00223-f010:**
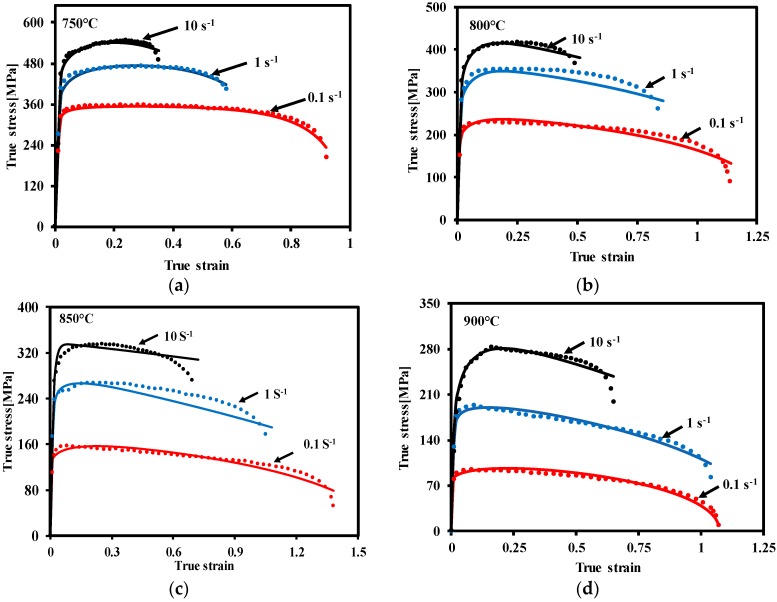
Comparison of stress-strain relationships of TA15 between experimental (symbols) and computed (solid curves) strain-stress at different hot stamping conditions, (**a**) 750 °C, (**b**) 800 °C, (**c**) 850 °C and (**d**) 900 °C. The strain rates used were 0.1 s^−1^, 1 s^−1^ and 10 s^−1^.

**Table 1 materials-12-00223-t001:** The main chemical alloying elements of TA15 (wt %).

Al	Zr	Mo	V	Ti
6.67	1.97	1.18	1.41	Remaining

**Table 2 materials-12-00223-t002:** Test matrix of the hot tensile tests.

Deformation Temperature (°C)	Strain Rates (s^−1^)
650, 700, 750, 800, 850, 900, 950	0.1, 1, 10

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
