# Peer review of "Deformation Behavior and Microstructural Evolution during Hot Stamping of TA15 Sheets: Experimentation and Modelling"

_materials, 2019, doi:10.3390/ma12020223_

Reviewer 1 Report

The paper is quite well organized and prepared, however some corrections are required in my opinion (listed below).

Tittle

I suggest to delete “Investigation of … ” and not to repeat “Hot”. Please consider to use following title: “Deformation Behavior and Microstructural Evolution during Hot Stamping of TA15 3 Sheets: Experimentation and Modelling”.

Abstract

Line 16 – “Scanning Electrical Microscope” term is unacceptable! It should be “Scanning Electron Microscope".

Keywords

Line 24 – The keyword “phase” is very general. I suggest to write “phase evolution”.

2.1. Materials and Experimental programme

Line 90 – “annealing tempere”? I suspected that the alloy was just annealed.

Line 97 – There is no “optical microstructure”! The light microscope is quite often wrongly called “optical microscope”. All microscopes have an optics. The caption should be written as follow: “Initial microstructure of TA15 alloy (light microscope)”.

Line 98 – What “the main chemical composition” does mean? I suspect you mean the “main alloying elements”. It can be just written “Chemical composition of …”. By the way, the content of elements was measured or given by the manufacturer? It should be written for the potential reader.

2.2. Equipments

Line 108 – I strongly suggest to use the term “thermocouple(s)” only – instead of “thermal couple(s)”.

Line 109 – see remark for line 16! Moreover, I suggest to simplify the first two sentenced – to avoid repetitions. “The microstructure observations was performed on S3400 Hitachi SEM microscope at Imperial College London.”

2.3. Test program

Lines 131-132 – You suggest the same mechanism of microstructural evolution in the temperature range of 650-950 deg. C. What is the phase transformation temperature range for examined titanium alloy? There is no showed in the manuscript any microstructure after deformation at 650 deg. C. What is the reason of that?

3. Material modelling

Line 180 – I suggest to use “thermomechanical processing” form instead of “thermo-mechanical” … or “thermos- mechanical” (lines 253-254).

4.2. Microstructure observation

Lines 274-275 – How do you distinguish “beta” phase from “transformed beta” phases? Even you don’t have any experimental evidences it can be assumed based on the phase transformation temperature range. The problem is there is no any information in the text about the start and finish temperature of alpha+beta->beta transformation – see previous remark for lines 131-132.

5. Conclusions - CRITICAL REMARKS

I. - lines 313-315 – What is the basis of the oxidation resistance evaluation?

II. – lines 316-320 – How did you evaluate the dislocation accumulation? How did you recognize dynamic recovery and recrystallization processes during hot deformation at selected temperatures? Is there any effect of alpha+beta->beta transformation on deformation behavior during hot stamping?

Presented conclusions are not supported by the results in my opinion. They should be reconsidered and rewrite.

Author Response

Dear Reviewer

Thanks very much for your comments for improving the paper. In terms of your comments, the revisions were made using track changes in the manuscript. In addition, individual response is explained and given below:

Title I suggest to delete “Investigation of …” and not to repeat “Hot”. Please consider to use following title “Deformation Behavior and Microstructural Evolution during Hot Stamping of TA15 sheets: Experimentation and modelling”

Response: The title has been changed, please see the revised manuscript.

Abstract Line 16 – “Scanning Electrical Microscope” term is unacceptable! It should be “Scanning Electron Microscope”

Response: Thanks for pointing out the mistake. The term has been changed in Line 16.

Keywords Line 24 – The key word “phase” is very general. I suggest to write “phase evolution”

Response: The keyword has been updated.

2.1 Materials and Experimental programme

Line 90 – “annealing tempered”? I suspected that the alloy was just annealed.

Response: The initial heat treatment temper of as-received is O condition provided by the material supplier, which is a stable condition. The previous term “annealing” is confusing and revised to “annealed”, please the revised Line 93.

Line 97 – There is no “optical microstructure”! The light microstructure is quite wrongly called “optical microscope”. All microscopes have an optics. The caption should be written as follows: “Initial microstructure of TA15 alloy (light microscope)”.

Response: We appreciate the professional comment. The term has been changed accordingly. Please see the revised line 97 in the manuscript.

Line 98 – What “the main chemical composition” does mean? I suspect you mean the “main alloying elements”. It can be just written “Chemical composition of …”. By the way, the content of elements was measured or given by the manufacturer? It should be written for the potential reader.

Response: We agree with the reviewer’s comments, chemical composition refers to the used main allying elements, which was given by the material supplier. We revised both the table caption and text. Please see the revised Lines 96-97 and Line 103.

2.2 Equipments

Line 108 – I strongly suggest to use the term “thermocouple(s)” only – instead of “thermal couple(s)”.

Response: Thanks. The term has been revised throughout the paper.

Line 109 – see remark for line 16! Moreover, I suggest to simplify the first two sentenced – to avoid repetitions. “The microstructure observations was performed on S3400 Hitachi SEM microscope at Imperial College London.”

Response: Thanks. The sentence has been simplified. Please see the revised Lines 115-116 in the manuscript.

2.3 Test program

Lines 131-132 – You suggest the same mechanism of microstructural evolution in the temperature range of 650-950 deg. C. What is the phase transformation temperature for examined titanium alloy? There is no showed in the manuscript any microstructural after deformation at 650 deg. C. What is the reason of that?

Response: For this type of titanium alloy, the temperature range of beta phase transformation is between 600 to 990 °C roughly. The manuscript has been revised with the temperature range of phase transformation added in Lines 132-133. The full transformation of beta phase is detrimental to ductility during hot deformation. Therefore, we controlled the temperature below that. Regarding the microstructure at 650 °C, the temperature is too low for ductility, which is not our interest from the view of hot stamping. We mainly concentrate on the macroscopic ductility performance first. Therefore, we did not do the microstructure for that temperature to minimize the work.

3 Material modelling

Line 180 – I suggest to use “thermomechanical processing” form instead of “thermo-mechanical”.. or “thermos-mechanical” (Lines 253-254).

Response: Thanks for the advice. The form has been changed.

Line 274-275 – How do you distinguish “beta” phase from “transformed beta” phases? Even you don’t have any experimental evidences it can be assumed based on the phase transformation temperature range. The problem is there is no any information in the text about the start and finish temperature of alpha+beta – beta transformation – see previous remark for lines 131-132.

Response: We are sorry for the missing of such an important information. The phase transformation temperature was added at necessary locations. Please also see the Lines 271 – 272 in the revised manuscript.

5 Conclusions -CRITICAL REMARKS

I. lines 313-315 – What is the basis of the oxidation resistance evaluation?

Response: Obvious oxidation were observed in the fracture morphology and specimen surface deformed at 950 °C in the experiments. At such a high temperature, with the proceeding of deformation, micro-cracks are easily initiated on the surface, severe oxidation initiated from the surface (failure of oxidation resistant lubricant) and penetrated, resulting in the quick fracture as observed in Figure 5(a). We add more explanations in Lines 236-240 in the revised manuscript.

II. – Lines 316 -320 – How did you evaluate the dislocation accumulation? How did you recognize dynamic recovery and recrystallization processes during hot deformation at selected temperatures? Is there any effect of alpha + beta – beta transformation on deformation behaviour during hot stamping.

Response: Thanks for the comments. Dislocation trend can be evaluated using the normalized dislocation density in our model. Strain hardening accumulated dislocations, while softening like recovery and recrystallization eliminated them. We focused on predictions of stress-strain behaviors. Therefore, we did not measure actual dislocation density, however our model enables to predict dislocation evolution trend and well predict macroscopic mechanical properties based on reasonable microstructural evolution.

Dynamic recovery is a very common microstructural feature for hot forming to eliminate accumulated dislocations. For TA15 titanium alloy, it can have two different crystal structures hcp  phase and bcc. The stacking fault energy of bcc  phase is very high, therefore, we assumed only dynamic recovery occurred for beta phase, while alpha phase occurred both recovery and recrystallisation. This is the effect of phase transformation during hot stamping. Within our temperature range, dynamic recrystallisation have been already identified, please see the added EBSD photos, Fig. 8 in Lines 297-298. This paper mainly concentrated on experimentation (stress-strain) and modelling. We value the reviewer’s comment and will put the detailed and systematic microstructure characterization research for future work.

Presented conclusions are not supported by the results in my opinion. The should be reconsidered and rewrite.

Response: Thanks. The conclusions have been rewritten with more links to the manuscript. The supporting statements, like transformation effect, oxidation resistance information and dynamic recrystallisation, were added in the revised manuscript to support our conclusions.

Best regards

Authors

Reviewer 2 Report

This paper is actual and could have been of interest for the scientific community. However, there are some issues must be clarified and revised in the manuscript and I recommend the editor to require a major revision.

I would like to highlight the following comments and issues:

1.     Line 90: Here, the authors say that the initial grain size is 3 micrometers. First, it is not clear to which phase this applies: alpha, betta, or the average size of both phases? Secondly, how did the authors measure the grain size? The grain boundaries are not visible in Figure 1.

2.     4.1. Hot tensile tests. It is not obvious why the authors attributed ductility decreasing with the failure of oxidation resistant lubricant. Authors need to specify the composition of the lubricant. Also, the authors say that it is due to the grain growth. I think that the analysis of the grain structure before hot deformation should be presented (annealing at the deformation temperatures).

3.     Lines 126-127. Here, the authors say, that: “The deformation temperatures were controlled below the upper limit of the ? phase transformation temperature.” Since the temperature range is selected in advance, why the authors did not indicate the polymorphic transformation temperature (β-transus)?

4.     Fig. 7. «Post-forming microstructure obtained for TA15…» Why is the post-forming? I think it should be “after deformation” or “after the tensile strain”.

5.     The authors say that their results may be used for real novel hot stamping process. However, this process is carried out in a cold dies, when the sheet will be cooled. In this paper, authors conducted tests on uniaxial tension at a constant temperature. How does this make it possible to talk about applicability to the actual stamping process?

6.     The authors conclude that the optimal temperature range corresponds to 750-900 °C. However, samples at deformation temperatures of 850–900 °C exhibit almost twice higher ductility than at 750 °C. Will this ductility value be sufficient for a real hot stamping process?

7.     Microstructural analysis is poorly presented. Considering the fact that the authors in the text of the paper discussed the microstructure evolution. However, they do not analyze the effect of temperature and strain rate on the final grain size. I think it is important, because this parameter has an effect on the final mechanical properties.

8.     Fig. 9c. Measurement units for the strain rate - must be small “s-1».

Author Response

Dear Reviewer

We appreciate your critical and professional comments for enhancing the quality of this paper. The revisions were made using track changes, please see the revised manuscript. Explanations of the responses are given below:

1. Line 90: Here, the author say that the initial grain size is 3 micrometers. First, it is not clear to which phase this applies: alpha, betta, or the average size of both phases? Secondly, how did the author measure the grain size? The grain boundaries are not visible in Figure 1.

Response: Thanks for the comment. The grain size refers to the average grain size of both alpha and beta phases, which is given by the material supplier. An additional EBSD photo of as-received condition was added to Fig. 8(a), please see the revised manuscript.

2. 4.1 Hot tensile tests. It is not obvious why the authors attributed ductility decreasing with the failure of oxidation resistant lubricant. Authors need to specify the composition of the lubricant. Also, the authors say that is due to grain growth. I think the analysis of the grain structure before hot deformation should be presented (annealing at the deformation temperatures).

Response: The reason of how the oxidation resistance lubricant affected the ductility is added. Please see Lines 236-240 in the revised manuscript. For the grain growth, we deleted this possible reason, although the grains tend to grow at high temperature, like 950 °C, the extent is not significant. Thanks for the reviewer’s critical comment to make the paper more clear.

3. Lines 126-127. Here, the authors say, that: “The deformation temperatures were controlled below the upper limit of the  phase transformation temperature.” Since the temperature range is selected in advance, why the authors did not indicate the polymorphic transformation temperature (-transus)?

Response: Thanks for the comment. The phase transformation temperature rage was added at relevant locations. Please see Lines 132-133 in the revised manuscript.

4. Fig. 7 <> Why is the post-forming? I think it should be “after deformation” or “after the tensile strain”

Response: We agree with the reviewer’s comment. The caption of Fig. 7 was revised to avoid confusion.

5. The authors say that their results may be used for real novel hot stamping process. However, this process is carried out in a cold dies, when the sheet will be cooled. In this paper, authors conducted tests on uniaxial tension at a constant temperature. How does this make it possible to talk about applicability to the actual stamping process?

Response: The uniaxial tensile tests were performed to obtain scientific indications and guides of deformation behaviors under different hot stamping condition. While for real forming process, the scenario is complex and difficult to simulate considering the temperatures vary at different locations of the sheet blank. There is no doubt that, there is a temperature range in real process. The process is believed to be capable if the isothermally obtained ductility within this range could satisfy the straining requirement of a particular part, the uniaxial information can still provide convincing and useful information for process designers.

6.The authors conclude that the optimal temperature range corresponds to 750-900 °C. However, samples at deformation temperatures of 850–900 °C exhibit almost twice higher ductility than at 750 °C. Will this ductility value be sufficient for a real hot stamping process?

Response: We value the author’s practical comment. We have to say that, ductility under uniaxial state can only provide some guides to the real stamping process. For sheet metal forming, besides ductility, hardening and stress-state also plays very important roles on the real forming. For 750 °C, although the ductility is less, the hardening is improved which contributes to obtaining uniform deformation and suppressing localized deformation. More hardening is beneficial for sheet metal forming. Therefore, we conclude this temperature is usable for certain components with less straining requirement.

7.Microstructural analysis is poorly presented. Considering the fact that the authors in the text of the paper discussed the microstructure evolution. However, they do not analyze the effect of temperature and strain rate on the final grain size. I think it is important, because this parameter has an effect on the final mechanical properties.

Response: Thanks for the comment. The objective of microstructure observations is to identify involved mechanism during hot stamping to establish the material model. Section 4.2 is used to evidence the used mechanisms of the model, including phase transformation and recrystallisation. Additional EBSD observations were added in Lines 284-289 and Fig. 8 to evidence the dynamic recrystallisation. We value the reviewer’s comment, and will systematically investigate the grain size evolution in the future.

8. Fig. 9c. Measurement units for the strain rate - must be small “s-1».

Response: This might be caused by the display error of Fig. 10(new manuscript). We double checked, and the unit of strain rate is “s-1” with “-1” in superscript.

Best regards

Authors

Reviewer 3 Report

it would be nice if the authors get acquainted with the works (for 3. Material modelling): M. L. Falk, J. S. Langer and L. Pechenik, for example "Thermal effects in the shear-transformation-zone theory of amorphous plasticity: Comparisons to metallic glass data" DOI: 10.1103/PhysRevE.70.011507.

45 line: it is necessary to paint and about the disadvantages of this method

66 line: we suggest avoiding the word "dislocation" because it is a type of defect and just a model

One of the important characteristics when stamping is the thickness of the finished product. It would be good if the authors provide these results.

Author Response

Dear Reviewer

We appreciate your comments for improving the paper. The revisions were made using track changes, please see the revised manuscript. In addition, the responses are given below:

It would be nice if the authors get acquainted with the works (for 3. Material modelling): M. L. Falk, J. S. Langer and L. Pechenik, for example "Thermal effects in the shear-transformation-zone theory of amorphous plasticity: Comparisons to metallic glass data" DOI: 10.1103/PhysRevE.70.011507..

Response: Thanks for the useful and interesting reference. The model in this study is dislocation-, physical mechanisms-based for metallic alloys. Currently, there is no direct link to the given reference, but with no doubt will be considered in the future work.

2. 45 line: it is necessary to paint and about the disadvantages of this method.

Response: The limitation and requirement of this process is added. Please see the Lines 52-54 in the revised manuscript.

3. 66 line: we suggest avoiding the word "dislocation" because it is a type of defect and just a model

Response: Thanks for the comment. The microstructure evolution has been renamed as “dislocation accumulation and annihilation”. Please see the Line 68 in the revised manuscript.

4. One of the important characteristics when stamping is the thickness of the finished product. It would be good if the authors provide these results.

Response: We agree with the reviewer’s comment that thickness is a very important characteristic for sheet forming processes. However, the performed study in this paper is a fundamental study of evaluating the feasibility of hot stamping TA15 sheets and modelling. The practical forming trials of this alloy will be performed in the future work. We have to say that, thickness of the part depends on the formed geometry and other process conditions, such as blankholding force, friction etc. A previous research has given some thickness information of Ti-6Al-4V using this novel process [1].

[1] Kopec, M.; Wang, K.; Politis, D. J.; Wang, Y.; Wang, L.; Lin, J. Formability and microstructure evolution mechanisms of Ti6Al4V alloy during a novel hot stamping process. Mater. Sci. Eng. A 2018, 719, 72–81, doi:10.1016/j.msea.2018.02.038.

Best regards

Authors

Reviewer 4 Report

Dear Authors,

I consider very interesting the topic of your research work.

The author should describe more in detail the novelty of the process.

The manuscript is well structured; however same aspects have to be reviewed.

The composition of the alloy TA15 have to be commented at list.

In the Material modelling section:

p.4. line 137, could you give one reference work if you talk about the literature?

p.5. line 140. Could you rewrite the paragraph? Its meaning is confusing.

In the Results and discussion section:

Could you describe in Figure 5 why the behaviour at 950º in the curve? Why did this phenomenon take place at this temperature?

The Figure 7, a), c) and e) are not necessary. In my opinion, these images could be eliminated.

Author Response

Dear Reviewer

We appreciate your comments for improving the paper. The revisions were made using track changes, please see the revised manuscript. In addition, the responses are replied as follows:

1. The composition of the alloy TA15 have to be commented at list.

Response: The chemical composition is given the Table 1 in Line 103. In addition, some information regarding this alloy composition is added in the text. Please see Lines 96-97 in the revised manuscript.

2. In the Material modelling section:

p.4. line 137, could you give one reference work if you talk about the literature?

Response: Thanks for the comment. References were added at the required location, Line 143 in the revised manuscript.

3. p.5. line 140. Could you rewrite the paragraph? Its meaning is confusing.

Response: We are sorry about the confusing writing. The paragraph has been rewritten.

4. In the Results and discussion section:

Could you describe in Figure 5 why the behaviour at 950º in the curve? Why did this phenomenon take place at this temperature?

Response: The behaviour at 950 ºC is believed to be mainly caused by the failure of oxidation. Please see the revised explanations in Lines 236 – 240 in the revised manuscript.

5. The Figure 7, a), c) and e) are not necessary. In my opinion, these images could be eliminated.

Response: Figures 7 a), c) and e) are obtained for a larger zone which can be used to compare to relatively small zones. Readers could have a better statistical view of the phases. For the completion of the paper, can we require to keep these figures. Thanks for the comment.

Best regards

Authors

Round  2

Reviewer 1 Report

I appreciate you have considered a large majority of my suggestions. Although the text of your manuscript is significantly improved I can’t find some answers for my questions.

1. You have added to the text the sentence (lines 132-133): “Considering the ? phase transformation temperature range is between 600 °C and 990 °C”. How that temperature range was it determined? If it is based on literature data you should cite relevant source.

Assuming that it is true and based on the scheme presented in Fig. 4, for every deformation conditions used in your research the transformed beta phase should be present in microstructure of examined alloy. In Fig. 7 transformed beta phase is present after deformation at 900 deg only. Could you comment this result? I asked you in my review “How do you distinguish “beta” from “transformed beta” phases?”

2. Conclusions are still not supported by presented results.

- Why do you suggest to avoid “severe beta phase transformation”? It is unclear for me.

 Additionally – Line 93 – You can’t replace the term “annealing temper” by “annealed temper”. I suggested in my review to write: “The initial alloy was annealed (or in annealed condition) with …”.

Author Response

Dear Reviewer

Thanks again for your comments! We are sorry for missing your questions in the first round. Please find our responses below:

1. You have added to the text the sentence (lines 132-133): “Considering the ? phase transformation temperature range is between 600 °C and 990 °C”. How that temperature range was it determined? If it is based on literature data you should cite relevant source.

Response: The lower limit of the temperature, 600 °C, is determined according to the ref [1], which evidences that titanium alloys in this study occurs phase transformation from alpha phase to beta phase. The upper limit of beta phase transformation was provided by the material supplier, which is same with the value in the ref [2]. The references were added in the revised manuscript in Line 132, same with previous references [9] and [11].

1.   Bai, Q.; Lin, J.; Dean, T. A.; Balint, D. S.; Gao, T.; Zhang, Z. Modelling of dominant softening mechanisms for Ti-6Al-4V in steady state hot forming conditions. Mater. Sci. Eng. A 2013, 559, 352–358, doi:10.1016/j.msea.2012.08.110.

2.   Fan, X. G.; Yang, H.; Gao, P. F. Prediction of constitutive behavior and microstructure evolution in hot deformation of TA15 titanium alloy. Mater. Des. 2013, 51, 34–42, doi:10.1016/j.matdes.2013.03.103.

 Assuming that it is true and based on the scheme presented in Fig. 4, for every deformation conditions used in your research the transformed beta phase should be present in microstructure of examined alloy. In Fig. 7 transformed beta phase is present after deformation at 900 deg only. Could you comment this result? I asked you in my review “How do you distinguish “beta” from “transformed beta” phases?”

Response: During the heating, some alpha phases can transform to beta phase. The titanium alloy in this study is a near- titanium alloy. The stable elements for beta phase are less, which results in beta phases transform to alpha phase again during cooling. The transformed alpha phases are the lamellar secondary alpha within the beta phase grain. The key of distinguishing beta phase and transformed beta phase is whether there are lamellar secondary alpha phases generated. When the temperatures are lower, lamellar secondary alpha phases have no time to form or the quantity is less, the conventional light microscope and SEM observations cannot identify. More explanations are given in Lines 272-277 in the revised manuscript.

2. Conclusions are still not supported by presented results.

- Why do you suggest to avoid “severe beta phase transformation”? It is unclear for me.

Response: According to the above phase transformation analysis, the greater amount of transformed beta phase during heating, can result in a greater amount of lamellar secondary alpha during cooling, which is the “severe beta phase transformation”. As is well know to all, equiaxed primary alpha exhibits better formability, fatigue resistance than the lamellar secondary alpha. Currently, titanium alloys are mainly used for the aircraft industry. The fatigue property is very component for aircraft components. Therefore, we aim to obtain equiaxed- or combined equiaxed + lamellar microstructures, rather than too much lamellar microstructure (the transformed beta). We added additional explanations in the conclusion.

 Additionally – Line 93 – You can’t replace the term “annealing temper” by “annealed temper”. I suggested in my review to write: “The initial alloy was annealed (or in annealed condition) with …”.

Response: Many thanks for the suggestion. The statement has been changed.

Reviewer 2 Report

The authors significantly revised the manuscript. Publication of the paper is recommended.

Author Response

 Dear Reviewer

 Many thanks for your approval and suggestion on our manuscript !

 Best regards

 The authors